# Seasonal and Dietary Effects on the Hematobiochemical Parameters of Creole Goats in the Peruvian Andes

**DOI:** 10.3390/vetsci12080687

**Published:** 2025-07-23

**Authors:** Aníbal Rodríguez-Vargas, Emmanuel Alexander Sessarego, Katherine Castañeda-Palomino, Huziel Ormachea, Fritz Trillo, Víctor Temoche-Socola, José Antonio Ruiz-Chamorro, Juancarlos Alejandro Cruz

**Affiliations:** 1Dirección de Supervisión y Monitoreo en las Estaciones Experimentales Agrarias, Instituto Nacional de Innovación Agraria (INIA), Lima 15024, Peru; e.sessarego14@gmail.com (E.A.S.); milyk12@gmail.com (K.C.-P.); trillozat@outlook.com (F.T.); jaruizch@gmail.com (J.A.R.-C.); jcruz@inia.gob.pe (J.A.C.); 2Estación Experimental Canaán, Instituto Nacional de Innovación Agraria (INIA), Ayacucho 05003, Peru; huziel.ormachea@gmail.com

**Keywords:** creole goats, blood biochemistry, hematological parameters, mixed models, goat nutrition

## Abstract

This study analyzed the effects of season and diet on the health of goats raised in the Peruvian Andes, where the climate is often extreme and variable. The study involved 45 young goats that received different diets during the rainy and dry seasons. Throughout the study, blood samples were collected, and body weight was monitored to assess physiological responses. Seasonal changes impacted health indicators more significantly than diet did. During the rainy season, goats showed better nutritional status, higher hemoglobin concentration, and lower immune response. In contrast, an increased immune response was observed in the dry season, possibly due to more challenging environmental conditions. Diet had a positive but limited effect, particularly on immune function. These findings are important for developing feeding and management strategies to improve animal health in high-Andean regions, thereby contributing to sustainability and food security for vulnerable communities.

## 1. Introduction

Goat production in the Peruvian Andes plays a crucial role in the economy and food security of rural communities, especially in regions such as Ayacucho. Creole goats have developed physiological adaptations that allow them to survive in extreme environments characterized by large temperature fluctuations, hypoxia, and variability in the availability of forage resources [1]. Despite those adaptations, extreme environmental events challenge their homeostasis, expressed in their hematological and biochemical profiles, impacting their health and productivity.

Several studies have shown that environmental factors such as altitude and seasonality significantly influence blood parameters in ruminants. In sheep and cattle exposed to hypoxia, increases in hematocrit and red blood cell count have been reported as compensatory mechanisms to optimize oxygen transport [2]. Likewise, heat stress and seasonal forage availability can modify the concentrations of blood metabolites and immune cells in goats, affecting their nutritional status and adaptive capacity [3,4].

Diet is another key factor in goat physiology. Previous research has shown that diet composition and strategic forage supplementation can modulate essential hematological parameters. For example, in Ecuador, the inclusion of *Moringa oleifera* in the diet of Creole goats promoted a more robust immune response, evidenced by an increase in leukocyte count [5]. However, the influence of specific nutritional strategies on goats adapted to Andean conditions remains underexplored.

To evaluate the impact of seasonality, diet, and sampling time on hematological, biochemical, and body weight parameters in goats, linear mixed models (LMMs) were used. These models allow simultaneous analysis of the fixed effects of environmental and dietary variables, as well as random effects associated with individual variations within the study population. In previous research, MLMs have proven their effectiveness in evaluating longitudinal data in ruminants, providing detailed information on physiological changes in response to environmental and nutritional factors [6].

Despite the importance of these factors, there is limited information on how the interaction between seasonality and diet affects the physiology of Creole goats in the Peruvian Andes. In this context, this study aimed to evaluate the effects of seasonal variability and diet on the hematological and biochemical parameters of Creole goats in Ayacucho, Peru, using linear mixed models to integrate fixed and random effects. The results of this research will contribute to the understanding of the physiological adaptations of these animals and provide key information for optimizing management strategies in high-altitude production systems.

## 2. Materials and Methods

### 2.1. Place and Duration of the Study

The study was conducted at the Lalaja Farm, located in the Sancos District, Lucanas Province, Peru, at coordinates −15.111152 latitude and −74.198218 longitude, at an altitude of 3100 m above sea level (Figure 1). This region has a temperate and dry climate, with an average annual temperature of approximately 10 °C, an average maximum temperature of 15 °C, and a minimum of 7 °C. The accumulated annual precipitation is around 261 mm, distributed over approximately 23 rainy days per year. The average wind speed is 10 km/h.

### 2.2. Study Animals and Treatment Assignment

The study was conducted with 45 Creole goats from the evaluation area, randomly assigned to three experimental groups of 15 animals each. Treatments were evaluated under two contrasting climatic conditions: rainy season (E1) and dry season (E2). Three feeding regimes were established: D1 (exclusive grazing), D2 (grazing with oat hay supplementation [54%] and alfalfa [46%]), and D3 (grazing with a concentrate supplement). Fifteen animals were evaluated in each season and diet combination, ensuring a balanced treatment allocation.

Environmental conditions showed substantial differences between seasons. During the rainy season (January–March), accumulated rainfall of up to 180 mm and an average temperature of 9 °C were recorded. In contrast, the dry season (June–August) saw rainfall of less than 30 mm and an average temperature of 6 °C. In both seasons, the average wind speed was 10 km/h. These climatic variations, characteristic of the dry temperate climate of the Andean region at 3,100 m above sea level, directly influence forage availability and the physiological and productive responses of animals (Figure 1).

### 2.3. Animal Handling and Experimental Conditions

The 12-month-old goats, with an average weight of 16 kg, were dewormed before the experiment. They were housed in separate pens and subjected to a week of adaptation to the pens or feedlots before the start of the 60-day trial. During this period, body weight, feed intake, and physiological response were monitored, following the protocol of [7]. The animals grazed extensively during the day and were confined in the afternoon, with ad libitum access to water.

### 2.4. Diets and Supplementation Protocols

The animals received two daily rations (8:00 and 17:00 h) according to three treatments: D1 (control), with exclusive access to freely available natural pastures; D2, with 2000 g per day per goat of a mixture of oat hay (CP ≈ 10%) and alfalfa (CP ≈ 19%); and D3, with 400 g of concentrated supplement, formulated to cover energy and protein requirements, especially during the dry season. The D3 dose represented approximately 1% of live weight, following nutritional recommendations for small ruminants.

To prevent digestive disorders and promote ruminal adaptation, D2 and D3 supplements were introduced progressively over a week, starting with 50% of the final dose. This protocol was designed to evaluate the influence of differentiated feeding strategies on the biochemical and hematological profiles of Creole goats, in the context of contrasting climatic conditions in the Peruvian Andes.

### 2.5. Blood Sampling and Collection

Beginning thirty days after the beginning of the experiment, blood was drawn every 15 days (sampling times, M1: 30 d, M2: 45 d and M3: 60 d) by jugular vein puncture between 6:00 and 8:00 a.m. Two 5 mL samples were obtained per goat: one with EDTA for hematological analysis and one without anticoagulant for biochemical analysis. The samples were labeled and transported in ice-filled thermal bags. The samples without anticoagulant were clotted at 37 °C, centrifuged at 3500 rpm for 10 min at 21 °C, and stored at −20 °C until analysis.

### 2.6. Biochemical and Hematological Analysis

Urea (BUN), total protein (TP), and albumin concentration (ALB) (mg/dL) were determined spectrophotometrically using commercial kits. Alkaline phosphatase (ALP) and alanine aminotransferase (ALT) activity were quantified by colorimetric methods in an automated biochemical analyzer [8]. Hematological parameters, including total white blood cell count (WBCL), leukocyte count (cells/µL), neutrophil (NeuP) and lymphocyte (LymP) percentages, hemoglobin concentration (HGB, g/dL), and mean corpuscular volume (MCV, fL), were assessed using an automated analyzer [9].

### 2.7. Monitoring of Body Weight

Body weight (kg) was recorded biweekly using a calibrated digital scale, under controlled conditions (prior fasting and level surface) to minimize errors. These data allowed for an assessment of productive response based on diet and climatic conditions, providing key information on the effects of supplementation and seasonality on animal growth. Direct weighing was used due to its greater precision and lower bias compared to subjective methods.

### 2.8. Statistical Analysis

A linear mixed model (MLM) with repeated measures was used to evaluate the effect of the climatic season, the type of diet, and the sampling time on biochemical and hematological parameters and body weight, with a significance level of α = 0.05. Normality and homogeneity of variance were verified using the Shapiro–Wilk and Levene tests, respectively. The design included 15 replicates per treatment and three factors: season (rainy, dry), diet (D1: grazing; D2: grazing + hay; D3: grazing + concentrate), and sampling time (M1, M2, M3).

The statistical model used was as follows:$$Y_{ijkl}=u+S_{i}+D_{j}+M_{k}+{\left(SD\right)}_{ij}+{\left(SM\right)}_{ik}+{\left(DM\right)}_{jk}+{\left(SDM\right)}_{ijk}+C_{l}+\varepsilon_{ijkl}$$
where Y*_ijkl_* represents the response measured in goat *l*, in season *i*, under diet *j* and at sampling time k; μ is the overall mean; S*_i_*, D*_j_*, and M*_k_* are the fixed effects of season, diet, and sampling time, respectively; (SD)*_ij_*, (SM)*_ik_*, (DM)*_jk_*, and (SDM)*_ijk_* represent the fixed second- and third-order interactions between the factors; C*_l_* is the random effect of goat *l*, assuming C_l_~N (0, σ2a); ε*_ijkl_* is the residual error, modeled with a correlation structure suitable for repeated measurements in each individual.

The model was fitted for each dependent variable, obtaining AIC, BIC, log-likelihood, and random-effects standard deviation values. Marginal hypothesis testing was performed using F tests and *p*-values were adjusted using the Benjamini–Hochberg method (FDR controlled).

Multiple comparisons were performed using the Tukey-Kramer test, appropriate for unequal sample sizes, due to its efficient control of Type I errors and robust estimates of contrasts, considering main and simple effects.

All analyses were performed in R (v.4.3.1) and RStudio (v.2024.12.0) [10], using the packages lme4 for model fitting and emmeans for marginal means and multiple comparisons.

## 3. Results

### 3.1. Evaluation of Model Fit

Table 1 compares models with and without interaction using AIC and BIC, assessing the trade-off between simplicity and fit. The log-likelihood and sigma are reported as indicators of the strength of fit. Marginal and conditional R^2^; values reflect the variance explained by the fixed effects and the full model. The optimal model was selected for its lower AIC and BIC, higher marginal R^2^, and lower sigma, avoiding overfitting and ensuring explanatory robustness.

(a)Model fitting for blood parameters

The models showed variability in their explanatory capacity. TP and BUN concentrations achieved moderate fits (R^2^ = 0.61 and 0.64), indicating the influence of factors not included. The ALB model showed the best performance (R^2^ = 0.68, AIC = 77.58, BIC = 147.77), combining accuracy and parsimony.

ALT and ALP models recorded similar values (R^2^ = 0.66), but ALP showed high residual dispersion (Sigma = 366.87), limiting its reliability. Although more complex, they were not more efficient than the ALB model. The random effects reflected greater intraindividual variability in ALP (SD = 353.74) compared to ALB (SD = 0.07) and TP (SD = 0.37) models, which showed more consistent responses.

(b)Model adjustment for hematological parameters and body weight

HGB and MCV models showed the best hematological fits (R^2^ = 0.72 and 0.73). In contrast, WBCL and NeuP showed less explanatory power (R^2^ = 0.39 and 0.52), highlighting possible unmodeled immunological factors.

The BW model was the most accurate (R^2^ = 0.91), highlighting its usefulness in production studies. MCV had the lowest AIC and BIC, indicating statistical efficiency. In contrast, WBCL, NeuP, and LymP presented greater complexity without proportional gains in accuracy.

Regarding random variability, WBCL and NeuP showed greater dispersion (SD = 8.9 and 0.068), associated with immune instability. HGB had low variability (SD = 1.53 and 0.33), while MCV and LymP recorded intermediate values (SD = 9.12 and 3.13). LW exhibited the lowest variability (SD = 0.012), confirming its robust fit.

### 3.2. Marginal Hypothesis Tests for Fixed Effects

The fixed effects of season, diet, and sampling time, and their interactions, on biochemical, hematological, and body weight parameters in goats were analyzed using linear mixed models. Significance was determined using F tests, adjusted by the Benjamini–Hochberg (BH) method.

(a)Blood parameters

According to Table 2, season significantly influenced BUN, TP, and ALB (*p* < 0.001), reflecting marked seasonal variations. In contrast, diet had no effect (*p* > 0.05), suggesting limited sensitivity to dietary changes. All parameters were affected by sampling time (*p* < 0.001), demonstrating physiological variability.

Season × sampling time (BUN, ALB; *p* < 0.001), season × diet (ALB; *p* < 0.01), and diet × sampling time (BUN; *p* < 0.05) interactions indicated combined physiological modulations. ALP responded to diet (*p* < 0.01) and ALT to season (*p* < 0.001). The season × diet interaction also affected ALP (*p* < 0.01), and season × sampling impacted ALT (*p* < 0.001), highlighting synergistic effects of environment and time.

The triple interaction (season × diet × sampling time) was not significant, suggesting that season and sampling are the main modulating factors.

(b)Hematological parameters and body weight

Table 3 shows that the season affected all parameters and body weight (*p* < 0.001), except for lymphocytes (*p* = 0.86). Diet only modified ALP (*p* < 0.01). Sampling influenced leukocytes, neutrophils, hemoglobin, hematocrit, MCV, and body weight (*p* < 0.01).

The season × diet interaction was significant for all parameters except lymphocytes (*p* > 0.05), as was season × sampling (*p* < 0.001), highlighting the importance of periodic monitoring. Diet × sampling affected leukocytes, neutrophils, hemoglobin, and body weight (*p* < 0.05). The three-way interaction was significant for hemoglobin and MCV (*p* < 0.001), but not for lymphocytes or body weight (*p* > 0.05), indicating greater stability of the latter in response to the factors evaluated.

### 3.3. Comparison of Means According to Climatic Season, Diet, and Sampling Time

Blood and hematological parameters and BW varied significantly (*p* < 0.05) according to the season, diet, sampling time, and their interactions (Table 4, Table 5, Table 6 and Table 7; Figure 1 and Figure 2).

(a)Blood parameters

Urea (mg/dL)

BUN was significantly higher in the rainy season (49.81 ± 0.75 mg/dL) than in the dry season (36.04 ± 0.77 mg/dL). There were no differences between D1 and D3, but D2 showed lower values. Sampling time influenced BUN, with the highest concentration in M3 (47.55 ± 0.96 mg/dL) and the lowest in M2 (37.90 ± 0.92 mg/dL).

In the rainy season, concentrations were homogeneous among diets, whereas in the dry season, D2 had the lowest values. In the season × time interaction, M3 had the highest concentration in the rainy season (61.87 ± 1.30 mg/dL), whereas in the dry season, M2 and M3 had the lowest values.

In the third-order interaction, urea remained high in the rainy diet (48–51 mg/dL), with no differences between diets, while in the dry diet it decreased, being the lowest in D2 (33.23 ± 1.40 mg/dL). In the rainy diet, M3 presented the highest values in D1 and D2 (57.33 ± 2.26 and 58.73 ± 2.26 mg/dL), while M1 and M2 were more homogeneous (44–45 mg/dL). In the dry diet, the lowest values were recorded in D1-M2 (27.33 ± 2.26 mg/dL) and D2-M3 (31.18 ± 2.26 mg/dL). D3 showed similar values between seasons, with lower concentrations in M2 and M3 in the dry season (31.00 ± 2.26 and 33.87 ± 2.26 mg/dL).

(b)Biochemical parameters

Total protein (g/dL)

TP concentration was higher in the rainy season (11.43 ± 0.12 g/dL) than in the dry season (9.90 ± 0.12 g/dL), with no differences between diets (10.44 ± 0.15–10.86 ± 0.15 g/dL). TP decreased over time, reaching its maximum value in M3 (11.56 ± 0.15 g/dL), followed by M1 (10.49 ± 0.15 g/dL) and M2 (9.94 ± 0.15 g/dL).

In the rainy season, D1 showed the highest concentration (11.91 ± 0.21 g/dL) compared to D2 (10.98 ± 0.21 g/dL) and D3 (11.41 ± 0.21 g/dL), with no differences between the latter (*p* > 0.05). In the dry season, the diets were homogeneous (D1: 9.82 ± 0.21, D2: 9.91 ± 0.21, D3: 9.96 ± 0.22 g/dL).

The season × sampling time interaction revealed that M3 in the rainy season had the highest TP (12.19 ± 0.21 g/dL), while M2 in the dry season had the lowest (9.15 ± 0.21 g/dL). In the diet × sampling time interaction, D1-M3 had the highest value (12.10 ± 0.25 g/dL) and D3-M2 the lowest (9.83 ± 0.25 g/dL). Finally, the rainy-season-D1-M3 had the highest value (13.19 ± 0.36 g/dL) and the dry-season-D1-M2 the lowest (8.95 ± 0.36 g/dL).

Albumin (g/dL)

Albumin (ALB) concentration was significantly higher during the rainy season (2.83 ± 0.02 g/dL) compared to the dry season (2.75 ± 0.02 g/dL). Sampling time also had a significant influence, with the highest value in M1 (3.12 ± 0.03 g/dL) and the lowest in M2 (2.64 ± 0.03 g/dL) and M3 (2.61 ± 0.03 g/dL). In the rainy season, D1 showed the highest concentration (2.95 ± 0.04 g/dL), while D2 and D3 presented similar values. In the dry season, D2 and D3 slightly exceeded D1, although without significant differences (*p* > 0.05). The season × time interaction revealed the highest value in M1–rainy combination (3.39 ± 0.04 g/dL) and the lowest in M3–rainy combination (2.52 ± 0.04 g/dL). In the diet × time combination, M1 presented the highest concentrations for all diets. In the triple interaction, the highest values corresponded to the M1–rainy combination in D3 and D1 (3.42 ± 0.06 g/dL), while the lowest ALB was observed in M2–rainy–D3 (2.43 ± 0.06 g/dL).

Alkaline phosphatase (U/L)

Alkaline phosphatase (ALP) activity did not show significant differences between seasons (dry: 606.72 ± 44.95 U/L; rainy: 533.23 ± 43.72 U/L). However, significant differences were observed between diets (*p* < 0.05), with D3 showing the highest activity (745.03 ± 54.92 U/L), followed by D2 (686.02 ± 54.32 U/L), while D1 showed the lowest activity (278.88 ± 53.65 U/L). Sampling time did not significantly influence the interaction between season and diet. D2 and D3 had the highest values in the dry season (819.92 ± 76.31 and 753.90 ± 76.31 U/L, respectively), while D1 was consistently low. In the rainy season, D3 (736.16 ± 75.42 U/L) and D2 (552.12 ± 76.31 U/L) maintained higher values than D1.

In the diet × sampling interaction, D3 peaked in M1 (842.50 ± 92.37 U/L), whereas D1 showed reduced enzyme activity at all times. The triple interaction showed the highest values in the dry season for D2 and D3 in M1 (955.73 ± 130.63 and 862.93 ± 130.63 U/L, respectively). The lowest values were observed in D1–M1–dry (143.07 ± 130.63 U/L) and D1–M3–rainy (338.93 ± 130.63 U/L), as shown in Figure 2.

Alanine aminotransferase (U/L)

ALT activity was significantly higher in the dry season (16.61 ± 0.33 U/L) than in the rainy season (14.79 ± 0.33 U/L). No significant differences were found between diets.

Regarding the sampling time, M1 presented the highest activity (18.24 ± 0.40 U/L), while M2 and M3 were similar (14.43 ± 0.40 and 14.41 ± 0.40 U/L, respectively).

In the dry season, D3 (17.22 ± 0.60 U/L) and D1 (16.93 ± 0.57 U/L) showed higher values than D2 (15.67 ± 0.57 U/L). In the rainy season, D1 was the highest (15.98 ± 0.57 U/L). The highest activity was recorded in D1–M1–dry (23.93 ± 0.98 U/L) and D3–M1–dry (23.13 ± 0.98 U/L), while the lowest value was observed in D3–M2–rainy (11.93 ± 0.98 U/L).

(c)Hematological parameters and body weight

Leukocytes (cells/µL)

Leukocytes were significantly higher in the dry season (13.31 ± 0.87 × 10^3^/µL) than in the rainy season (8.06 ± 0.90 × 10^3^/µL), suggesting greater immune activation by environmental stress and lower forage availability.

Diet also had a significant effect (*p* < 0.05): D2 had the highest value (13.70 ± 1.01 × 10^3^/µL), followed by D3 (10.95 ± 1.19) and D1 (7.41 ± 1.05), indicating a possible immunomodulatory effect of the diet ingredients.

Regarding time, M3 recorded the highest count (13.63 ± 1.03), while M1 and M2 showed lower values (8.67 ± 1.17 and 9.76 ± 1.05, respectively).

During the dry season, D2 reached the highest peak (18.89 ± 1.48), while in the rainy season, D1 and D3 had the lowest values (~7.38–8.52), reflecting greater immune demand under heat stress.

The highest value was observed in M3–dry (18.67 ± 1.54), and the lowest (~7.65–8.59) in M1– and M2–rainy, reaffirming the effect of the environment on the immune response.

There were no significant differences in the triple interaction (*p* > 0.05), except in dry–D2–M3 (33.88 ± 2.77, the highest value) and dry–D1–M2 (5.80 ± 3.25, the lowest), suggesting greater immune activation under conditions of greater environmental stress and better dietary quality.

Neutrophils (%)

NeuP values did not show significant differences between seasons, diets, or sampling times (*p* > 0.05), suggesting a stable immune response. Although not significant, D1 (42.70 ± 2.80%), and D2 (41.33 ± 2.76%) exhibited higher mean concentrations than D3 (36.54 ± 2.93%). Similarly, NeuP showed a reduction trend from M1 (42.43 ± 2.20%) to M2 (36.83 ± 2.05%), suggesting potential variability in immune activation.

Hemoglobin (g/dL)

Hemoglobin (HGB) was higher in the rainy season (9.84 ± 0.16 g/dL) than in the dry season (8.86 ± 0.15 g/dL; *p* < 0.05). There were no differences between diets: D2 (9.62 ± 0.22 g/dL), D3 (9.33 ± 0.23 g/dL), and D1 (9.10 ± 0.22 g/dL; *p* > 0.05).

Sampling time had a significant influence, with decreasing values from M1 (10.77 ± 0.19 g/dL) to M3 (8.43 ± 0.16 g/dL; *p* < 0.05). The season × diet interaction showed higher values in the rainy season with D2 (10.50 ± 0.25 g/dL) and lower values in the dry season with D2 (8.74 ± 0.25 g/dL; *p* < 0.05).

In the season × time interaction, HGB was highest in the dry season during M1 (11.41 ± 0.21 g/dL) and lowest in M3 (7.49 ± 0.21 g/dL; *p* < 0.05). In diet × time, D2 in M1 reached the maximum (11.43 ± 0.28 g/dL) and in M3 the minimum (7.89 ± 0.28 g/dL; *p* < 0.05).

The triple interaction reflected the combined effect of environment and nutrition: the highest HGB was recorded in the dry season with D2 in M1 (11.99 ± 0.36 g/dL) and the lowest in the dry season with D2 in M3 (5.67 ± 0.36 g/dL; *p* < 0.05), evidencing the environmental and dietary impact on blood oxygenation.

Mean corpuscle volume (fL)

MCV varied significantly by season, with higher values in the dry season (18.51 ± 0.06 fL) than in the rainy season (18.34 ± 0.06 fL). However, diet had no significant effect, with similar values between D2 (18.56 ± 0.09 fL), D1 (18.45 ± 0.09 fL), and D3 (18.26 ± 0.09 fL).

Sampling time had a significant influence: M3 (18.59 ± 0.06 fL) and M2 (18.50 ± 0.06 fL) recorded higher values than M1 (18.18 ± 0.06 fL). In the season × diet interaction, D2 in the dry season reached the highest value (18.70 ± 0.10 fL), while D3 in the rainy season had the lowest (18.12 ± 0.11 fL).

In the season × sampling interaction, M3 and M2 in the dry season showed the highest values (18.73 ± 0.07 and 18.66 ± 0.07 fL), while M1 in the same season recorded the lowest (18.13 ± 0.07 fL). For the diet × sampling interaction, D2 in M3 presented the highest value (18.82 ± 0.10 fL) and D3 in M1 the lowest (18.06 ± 0.12 fL).

Finally, in the triple interaction, D2 in M3 during the dry season had the highest MCV (19.18 ± 0.12 fL), while D3 in M1 during the same season had the lowest (18.05 ± 0.12 fL). These results show that season and sampling time significantly affect MCV, whereas diet does not generate marked differences.

Lymphocytes (%)

LymP did not vary between seasons (*p* > 0.05), with similar values in the dry season (43.19 ± 1.68%) and rainy season (42.84 ± 1.72%). There were also no differences by diet (*p* > 0.05), although D3 had the highest LymP (47.37 ± 2.68%), followed by D2 (41.68 ± 2.55%) and D1 (39.99 ± 2.57%), as shown in Figure 3.

The sampling time influenced the values: M2 recorded the highest (45.72 ± 1.84%), while M3 (42.19 ± 1.82%) and M1 (41.13 ± 1.96%) were lower (*p* > 0.05). In the season × diet interaction, D3 in the dry season reached the highest LymP (48.83 ± 2.86%), while D1 in the rainy season had the lowest (39.88 ± 2.79%).

In the season × sampling interaction, M2 in the rainy season showed the highest value (47.54 ± 2.13%), while M3 in the same season recorded the lowest (40.48 ± 2.13%). For the diet × sampling time interaction, D3 in M2 presented the highest LymP (50.26 ± 3.07%), while D1 in M3 had the lowest (38.02 ± 3.11%).

In the triple interaction, D3 in M2 in the rainy season reached the highest LymP (50.45 ± 3.69%), while D1 in M3 in the same season recorded the lowest (34.97 ± 3.69%).

Body weight (kg)

BW was higher in the dry season (27.02 ± 0.48 kg) than in the rainy season (24.86 ± 0.48 kg; *p* < 0.05), with a reduction of 2.16 kg under more humid conditions. Diet did not have a significant influence (*p* > 0.05), although D3 showed the highest value (27.18 ± 0.82 kg), followed by D1 (26.05 ± 0.82 kg) and D2 (24.60 ± 0.82 kg), with a difference of up to 2.58 kg between treatments.

In the season × diet interaction, D3 in the dry season reached the highest weight (29.05 ± 0.83 kg), exceeding D2 in the rainy season (24.13 ± 0.83 kg) by 4.92 kg, which registered the lowest value. In the dry season, D1 (26.95 ± 0.83 kg) showed no differences with D3 (25.32 ± 0.83 kg). The same way, D1 was similar between season.

## 4. Discussion

### 4.1. Interpretation of Model

Model accuracy varied by parameter type, reflecting differences in biological stability and sensitivity to unobserved factors. The best fits were associated with variables with low intraindividual variability, such as ALB and red blood cell indices, supporting their usefulness as robust predictors in high-Andean systems.

In contrast, models related to immunological parameters (WBCL, NeuP) showed less explanatory power, possibly due to high individual variability or the absence of critical covariates, such as parasite load or vaccination history. This highlights the need to integrate more sensitive immunological biomarkers and multivariate approaches that capture the complexity of the immune system.

The live weight (LW) model was the most accurate, demonstrating its potential as a management tool in production systems with limited availability of longitudinal data.

The quality of the fit depends not only on the statistical technique, but also on the biological behavior of each variable and its interaction with the environment. Mixed models effectively captured fixed and random effects in hierarchical and repeated structures, proving key to animal husbandry modeling.

These results coincide with studies that report the influence of diet, season, and physiological state on goat biochemical and hematological profiles [4,11,12,13] and confirm the usefulness of these models for designing evidence-based management strategies [14]. These findings are consistent with studies reporting variations associated with diet, season, and physiological condition [4,11,12,13]. The use of mixed models allowed for the discrimination of fixed and random effects in hierarchical and repeated structures, which is key in zootechnical research [14].

### 4.2. Interpretation of Marginal Hypothesis Tests for Fixed Effects

The significant influence of season, sampling time, and, to a lesser extent, diet, highlights the sensitivity of physiological parameters to environmental factors and management practices. The observed interactions demonstrate non-additive biological responses, highlighting the adaptive complexity of goats in the face of seasonal and nutritional variations.

These multifactorial responses reflect homeostatic adjustments mediated by the environment and the diet, consistent with the recognized physiological plasticity of small ruminants to climatic and dietary stimuli [4,11,12,13,15,16].

Incorporating these effects through mixed models provides a robust and contextualized approach for the analysis of physiological determinants in variable production systems and underscores the importance of considering the interaction between ecological and operational factors when interpreting zootechnical data.

### 4.3. Comparison of Means According to Climatic Season, Type of Diet, and Sampling Time

(a)Biochemical parameters

Urea (mg/dL)

BUN levels were significantly higher during the rainy season (*p* < 0.001), in agreement with [17] who reported seasonal fluctuations in this metabolite. Ref. [18] attributed variations in urea to age, linked to changes in protein metabolism and kidney function. Ref. [9] reported lower urea concentrations in winter and elevated values during gestation (*p* < 0.05), which agrees with our findings of lower BUN in the dry season, possibly reflecting lower protein availability or higher nitrogen use efficiency.

On the other hand, Ref. [12] showed a decrease in BUN in goats fed Moringa oleifera (*p* < 0.05), an effect similar to that observed in our research for diet D2 during the dry season. In contrast, Ref. [13] did not find significant differences between dietary treatments (*p* > 0.05), which reinforces the multifactorial nature of urea metabolism.

Total protein (g/dL)

Seasonal variations in TP levels demonstrate their sensitivity to environmental conditions in high-Andean Creole goats. The increase observed during the rainy season (11.43 ± 0.12 g/dL) suggests greater forage availability and quality, which would favor hepatic plasma protein synthesis and a positive nitrogen balance.

This behavior coincides with that described by [4], who found maximum TP values in summer (68.75 mg/dL) and minimum values in winter (*p* < 0.05), reflecting the interaction between reproductive physiology and seasonality. The absence of significant differences between extensive and semi-intensive systems suggests that, under high-Andean conditions, climate could have a greater impact on TP than moderate dietary variations, as also indicated by [15,19], who found no effects of sex or diet type (*p* > 0.05).

However, other studies report the influence of physiological and genetic factors. In Northern Nigeria [20], identified differences according to breed, age, and sex (*p* < 0.05), while in Egypt. Ref. [21], reported values between 6.0 and 6.98 g/dL in intensive systems, attributed to management. In Nigeria [22] found lower concentrations in goats fed Andropogon gayanus compared to mixed diets (*p* < 0.05), highlighting the role of forage quality.

During the peripartum period, in Turkey [23] reported stable TP levels (3.5–13 g/dL), suggesting homeostatic mechanisms. In Nigeria [24] documented seasonal variations in Sahel goats: 5.83 ± 0.26 g/dL in the dry season, 6.54 ± 0.34 g/dL in the rainy season, and 6.48 ± 0.21 g/dL in the cold season.

In high-Andean ecosystems, where climatic fluctuations are more pronounced, these factors appear to have a greater impact on protein homeostasis than dietary adjustments, highlighting the need for nutritional strategies adapted to extreme ecological contexts.

Albumin (g/dL)

The concentration of ALB was significantly higher during the rainy season, suggesting the positive influence of more favorable environmental conditions on the protein metabolism and the nutritional status of the animals. This pattern is consistent with previous studies reporting seasonal stability in ALB, attributing such stability to a lower incidence of heat stress under moderate environmental conditions [25].

However, the interaction between season and sampling showed significant differences in ALB levels, reflecting physiological variability associated with specific temporal changes within each season.

Similar results were reported by [26], who observed a higher ALB concentration during the rainy season (4.93 g/dL). In the triple interaction, the M1 combination during the rainy season with days D3 and D1 presented the highest ALB values (3.42 ± 0.06 g/dL), while M2 in the rainy season with D3 showed the lowest levels (2.43 ± 0.06 g/dL), coinciding with [27], who recorded a significant reduction in ALB in the peripartum period (*p* < 0.05).

On the other hand, Ref. [12] reported a decrease in ALB levels after Moringa oleifera supplementation (*p* < 0.01); however, in the present study, diet had no significant effect on ALB (*p* > 0.05). Finally, Ref. [18] found a significant effect of age on albumin levels (*p* < 0.01), suggesting complex regulation involving physiological and environmental factors.

Alkaline phosphatase (U/L)

ALP levels did not vary between seasons, but did vary between diets, with significantly higher concentrations in D2 and D3 compared to D1 (*p* < 0.05), in line with [28], who linked them with greater osteogenic activity in young animals.

During the dry season, D2 and D3 showed higher values than D1, a pattern that was repeated during the rainy season. The M1 group showed the highest levels in the dry season. In the diet × sampling interaction, D3-M1 and D2-M1 stood out with the highest values; and in the triple interaction (diet × sampling × season), D2-M1 and D3-M1 were higher in the dry season (*p* < 0.05), while D1 maintained the lowest levels.

These results reinforce the relationship between ALP and bone metabolism, suggesting that D2 and D3 promote osteoblastic activity, possibly due to improved nutrient availability. However, Ref. [19] reported higher values in adults than in young individuals (*p* < 0.05), indicating an age effect on their regulation.

Alanine aminotransferase (U/L)

ALT activity was significantly higher in the dry season (*p* < 0.05), with no differences by diet (*p* > 0.05). The M1 group had the highest values, reflecting temporal variability. In the dry season, D3 and D1 exceeded D2, while in the rainy season, D1 predominated. The three-way interaction revealed maximum values in D1 and D3-M1 during the dry season, and in D1-M3 during the rainy season.

These results are consistent with [12], who linked ALT with dietary modifications, and with [29], who reported increases under similar conditions. Ref. [15] observed no effect of diet or sex (*p* > 0.05). Ref. [28] reported higher levels in adults than in young individuals, suggesting the influence of age.

(b)Hematological parameters and body weight

Leukocytes (cells/µL)

WBC concentrations were significantly higher in the dry season (*p* < 0.05), reflecting environmental stress-induced immune activation [30]. D2 presented the highest values, followed by D3 and D1, indicating a dietary effect [22]. Ref. [21] reported similar values in lactating goats (8.05–12.88 × 10^3^/μL). The highest concentration was observed in M3, while M1 and M2 were lower (*p* < 0.05), in line with [31]. Although other studies reported no differences [15,31], the highest value (33.88 × 10^3^/μL) was reached in dry–D2–M3, suggesting greater immune demand due to heat stress. Age and breed influenced WBC [32,33,34], although Ref. [18] did not observe age differences. Ref. [20] found higher values in adult Borno White goats and female kids. Ref. [35] reported a mean of 14.6 ± 3.32 × 10^3^/μL.

Neutrophils (%)

Values were stable across seasons, diets, and sampling times (*p* > 0.05), indicating immune stability. Ref. [30] reported segmented counts of 0.95–4.98 × 10^9^/L and band counts of 0–0.16 × 10^9^/L, influenced by environment and diet. Ref. [15] observed an effect of legume diets (*p* < 0.05), while Ref. [15] reported higher percentages in T2 (10.50%) compared to T1 (5.47%) and T3 (4.58%), in contrast to this study.

Refs. [5,13,31] found no differences in neutrophil values based on treatments or season (*p* > 0.05). Ref. [36] also found no seasonal variations, while Ref. [17] reported an increase in winter (*p* < 0.001).

Ref. [33] recorded higher values in CH goats compared to other breeds (*p* < 0.0001) and in males compared to females (*p* < 0.05). Ref. [37] reported an increase with age, without attributing it exclusively to this variable. In this study, these factors did not influence the effect. Ref. [38] reported lower values in Red Sokoto, while in Sahel there were no differences (*p* > 0.05), suggesting a breed–environment interaction not assessed in this study.

Hemoglobin (g/dL)

HGB values were higher in the rainy season (9.84 ± 0.16) than in the dry season (8.86 ± 0.15; *p* < 0.05), suggesting better oxygenation [17,39]. A progressive reduction was observed from M1 (10.77 ± 0.19) to M3 (8.43 ± 0.16; *p* < 0.05), in line with [5]. The lowest value was recorded in dry–D2 (8.74 ± 0.25) and the highest in rainy–D2 (10.50 ± 0.25; *p* < 0.05). The triple interaction revealed marked differences between M1 (11.99 ± 0.36) and M3 (5.67 ± 0.36) in dry–D2 (*p* < 0.05), indicating a strong climate–diet–production level interaction [12].

Ref. [40] reported higher levels in Moxotó (10.15) and Anglo-Nubian (9.38) breeds compared to Boer and Savana, demonstrating genetic differences in adaptation. The observed variations could be related to alterations in the ruminal microbiota and vitamin B12 synthesis, associated with protein and hematopoietic metabolism [41].

Mean corpuscle volume (fL)

The MCV was significantly higher during the dry season, with no effects attributable to diet type. The variations observed between specific diet and sampling combinations, especially in this season, suggest an influence of environmental and temporal factors on red cell morphology. The triple interaction demonstrated the sensitivity of MCV to seasonal conditions, possibly as a physiological response to changes in temperature, humidity, or nutrient availability.

Several studies support the variability of MCV depending on breed, age, and physiological status. Increases were observed in Saanen females [42], as well as differences between breeds such as Argentata dell’Etna [32], Aspromontana, Girgentana [34], and Kano Brown [20]. Ref. [18] also reported age-related variations in Girgentana, and [43] in Batina neonates. In the Sahel breed, kids showed higher values than in the Red Sokoto in the hot–dry season [44].

The environment also influences MCV, with higher values in winter [36] and in advanced gestation in Baladi goats [45]. Regarding diet, no significant effects were observed [14,46]. On the other hand, higher levels were reported in twin pregnancies compared to single pregnancies [38], which underscores the combined influence of physiological, environmental, and genetic factors on the regulation of MCV in goats.

Lymphocytes (%)

The proportion of lymphocytes remained stable across seasons and diets, indicating a low sensitivity of this leukocyte subpopulation to environmental and nutritional variations. However, trends toward higher values were observed in animals on the D3 diet and during the second sampling, suggesting possible immunological modulation by specific factors not yet identified.

These results are consistent with those of [5,13,31], who found no differences between treatments or physiological states. Ref. [36] also reported seasonal stability, while Ref. [30] recorded variability based on diet and environment.

In India Ref. [17] reported lower LymP in females in winter (*p* < 0.01), and in Nigeria [22] reported lower LymP with Pterocarpus erinaceus supplementation (*p* < 0.05). Ref. [33] observed a reduction with age (*p* < 0.0001) and higher values in females (*p* < 0.05), and [37] reported a shift from lymphocytic to neutrophilic predominance. Ref. [44] reported increased LymP in Red Sokoto (*p* < 0.05), but no differences were observed in Sahel (*p* > 0.05). LymP shows relative stability, although it is influenced by age, breed, and physiology.

Body weight (kg)

Body weight was higher in the dry season, indicating more favorable environmental conditions for growth. Although diet alone did not show significant effects, D3 was associated with higher weights. The season × diet interaction showed better performance with D3 in the dry season and lower performance with D2 in the rainy season, suggesting that growth is modulated by a combination of nutritional and seasonal factors, possibly linked to forage availability and heat stress. These findings coincide with the findings of [47], who linked weight with energy intake (r = 0.87, *p* < 0.001). Previous studies associate weight loss with heat stress and lower consumption [48,49,50], reporting growth compensation after weaning.

## 5. Conclusions

Predictive models applied to the analysis of blood, hematological, and body weight parameters allow for the identification of complex interactions between environmental and nutritional factors. These tools are key to optimizing feeding and management strategies, improving production efficiency and resilience in tropical systems.

The season, diet, and time after dietary intervention significantly influence blood and hematological parameters and body weight in Creole goats, demonstrating complex interactions. The rainy season favors metabolic status and blood oxygenation, possibly due to greater nutrient availability, while the dry season conditions promote immune response and productive performance.

Adjusting diets to environmental conditions is essential to mitigate the impact of cli-mate stress on tropical livestock production. An adequate protein balance optimizes metabolic and immune responses, promoting greater productive efficiency and system sustainability.

## Figures and Tables

**Figure 1 vetsci-12-00687-f001:**
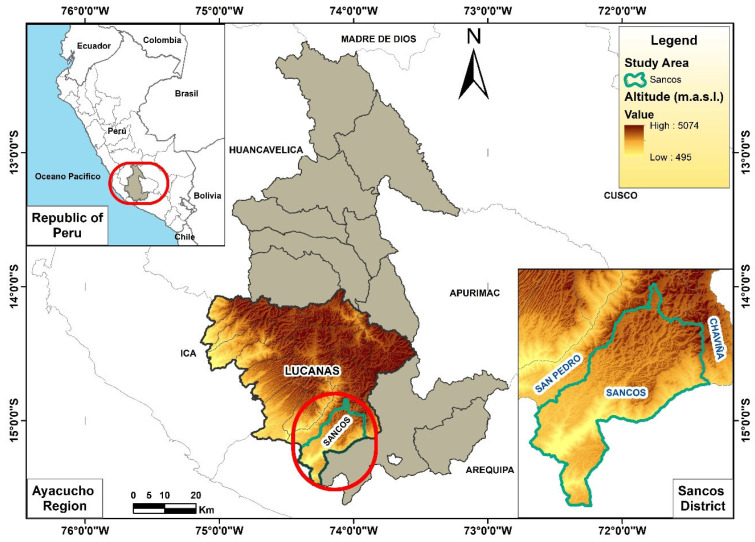
Geographic location of the study area.

**Figure 2 vetsci-12-00687-f002:**
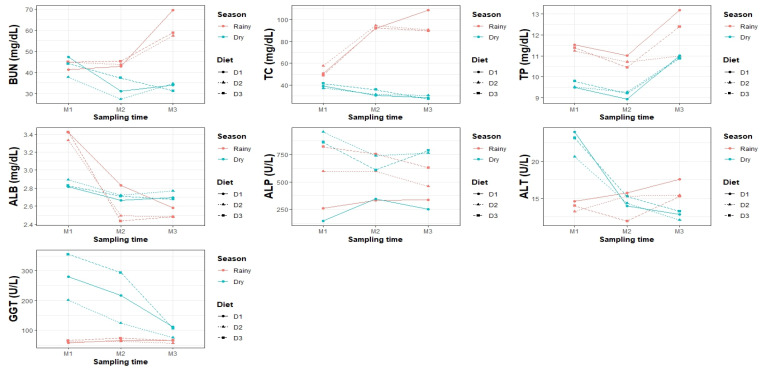
Interaction between climatic season, diet, and sampling time regarding blood parameters in goats.

**Figure 3 vetsci-12-00687-f003:**
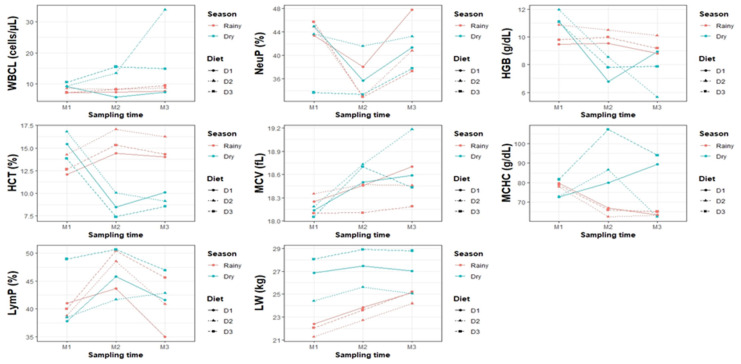
Interaction between climatic season, diet, and sampling time regarding hematological parameters and body weight of goats.

**Table 1 vetsci-12-00687-t001:** Measures of model fit for blood, hematological, and body weight parameters in goats.

Blood Parameters							Random Effect (Goat)
Variable	*n*	AIC	BIC	Log-Like	Sigma	R^2^	SD
Urea (BUN)	265	1858.79	1928.97	−909.39	8.47	0.64	2.19
Protein (TP)	265	945.02	1015.21	−452.51	0.41	0.61	0.37
Albumin (ALB)	265	77.58	147.77	−18.79	0.23	0.68	0.07
ALP	262	3740.14	3810.08	−1850.07	366.87	0.66	353.74
ALT	265	1429.60	1499.79	−694.80	3.37	0.66	1.76
Hematological parameters and body weight							
Leukocytes (WBCL)	240	1694.54	1762.50	−827.27	9.21	0.39	0.012
Neutrophils (NeuP)	240	1907.75	1975.71	−933.87	13.31	0.52	8.9
Hemoglobin (HGB)	256	891.06	960.42	−425.53	1.20	0.72	068
MCV	256	285.95	355.31	−122.97	0.31	0.73	0.33
Lymphocytes (LymP)	240	1846.71	1914.67	−903.35	11.44	0.56	8.44
Body weight (BW)	270	1077.41	1147.92	−518.71	1.28	0.91	3.13

*n*: sample size; ALP: alkaline phosphatase; ALT: alanine aminotransferase; MCV: mean corpuscular volume; AIC: Akaike Information Criterion; BIC: Bayesian Information Criterion; Log-Like: log-likelihood; Sigma: standard deviation of the residuals; R^2^: coefficient of determination; SD: standard deviation.

**Table 2 vetsci-12-00687-t002:** Marginal hypothesis tests of blood parameters.

Effect	GL	Urea (BUN)		Protein (TP)		Albumin (ALB)	
	DF	F-Value	*p*-Value BH	F-Value	*p*-Value BH	F-Value	*p*-Value BH
(Intercept)		4861.78 ***	0.00	11,608.27 ***	0.00	26,385.61 ***	0.00
Season	1	174.72 ***	0.00	88.40 ***	0.00	7.70 **	0.00
Diet	2	2.80	0.08	1.55	0.29	1.71	0.20
Sampling time	2	28.13 ***	0.00	33.26 ***	0.00	135.12 ***	0.00
Season × Diet	2	0.77	0.46	3.37	0.07	6.97 **	0.00
Season × Sampling time	2	60.05 ***	0.00	0.86	0.42	70.29 ***	0.00
Diet × Sampling time	4	3.19 *	0.01	1.83	0.19	1.50	0.20
Season × Diet × Sampling time	4	4.90 **	0.00	1.23	0.34	1.90	0.15
		**ALP**		**ALT**			
(Intercept)		95.15 ***	0.00	2205.09 ***	0.00		
Season	1	1.90	0.27	19.53 ***	0.00		
Diet	2	5.86 **	0.01	1.37	0.26		
Sampling time	2	1.47	0.30	37.87 ***	0.00		
Season × Diet	2	5.43 **	0.01	4.19 *	0.02		
Season × Sampling time	2	0.46	0.71	72.50 ***	0.00		
Diet × Sampling time	4	2.10	0.16	1.61	0.23		
Season × Diet × Sampling time	4	0.53	0.71	1.36	0.26		

DF: degrees of freedom; ALP: alkaline phosphatase; ALT: alanine aminotransferase; *p*-value BH: *p*-value adjusted by the Benjamini–Hochberg method. F-values are expressed with superscripts according to the level of significance: *** *p* < 0.001; ** *p* < 0.01; * *p* < 0.05. Season: refers to the seasons (rainy season and dry season).

**Table 3 vetsci-12-00687-t003:** Significance analysis of marginal hypothesis tests for hematological parameters and body weight in goats.

Effect	DF	Leukocytes (WBCL)		Neutrophils (NeuP)		Hemoglobin (HGB)	
		F-Value	*p*-Value BH	F-Value	*p*-Value BH	F-Value	*p*-Value BH
(Intercept)	1	288.91 ***	0.00	288.91 ***	0.00	5240.00 ***	0.00
Season	1	17.20 ***	0.00	17.20 ***	0.00	37.55 ***	0.00
Diet	2	9.33 ***	0.00	9.33 ***	0.00	1.43	0.25
Sampling time	2	5.98 **	0.00	5.98 **	0.00	71.81 ***	0.00
Season × Diet	2	6.25 **	0.00	6.25 **	0.00	7.67 **	0.00
Season × Sampling time	2	3.83 *	0.02	3.83 *	0.02	44.44 ***	0.00
Diet × Sampling time	4	4.43 **	0.00	4.43 **	0.00	8.99 ***	0.00
Season × Diet × Sampling time	4	4.90 **	0.00	4.90 **	0.00	10.76 ***	0.00
		**MCV**		**Lymphocytes (LymP)**		**Body Weight (BW)**	
(Intercept)	1	117,670.15 ***	0.00	835.21 ***	0.00	2999.70 ***	0.00
Season	1	16.38 ***	0.00	0.08	0.86	187.43 ***	0.00
Diet	2	2.71	0.07	2.10	0.27	2.53	0.10
Sampling time	2	34.20 ***	0.00	3.12	0.18	20.46 ***	0.00
Season × Diet	2	7.31 ***	0.00	0.90	0.65	27.05 ***	0.00
Season × Sampling time	2	8.92 ***	0.00	2.11	0.27	43.09 ***	0.00
Diet × Sampling time	4	2.26	0.07	0.32	0.86	2.85 *	0.03
Season × Diet × Sampling time	4	5.79 ***	0.00	0.58	0.86	0.23	0.92

DF: degrees of freedom; MCV: mean corpuscular volume; BH: Benjamini–Hochberg-adjusted *p*-value. F-values are expressed with superscripts according to the level of significance: *** *p* < 0.001; ** *p* < 0.01; * *p* < 0.05.

**Table 4 vetsci-12-00687-t004:** Goats’ blood biochemistry by season and diet (values adjusted with *p*-value BH).

Season	Diet	Urea	Protein	Albumin	ALP	ALT
		(mg/dL)	(g/dL)	(g/dL)	(U/L)	(U/L)
Rainy	D1	51.24 ^a^	11.91 ^a^	2.95 ^a^	311.42 ^c^	15.98 ^ab^
	D2	48.60 ^a^	10.98 ^b^	2.77 ^b^	552.12 ^b^	14.64 ^bc^
	D3	49.60 ^a^	11.41 ^ab^	2.78 ^b^	736.16 ^ab^	13.73 ^c^
Dry	D1	37.38 ^b^	9.82 ^c^	2.72 ^b^	246.34 ^c^	16.93 ^a^
	D2	33.24 ^c^	9.91 ^c^	2.79 ^b^	819.92 ^a^	15.67 ^ab^
	D3	37.17 ^b^	9.96 ^c^	2.74 ^b^	753.90 ^ab^	17.22 ^a^
Standard error		1.27	0.21	0.04	75.42	0.57

D1: exclusive grazing; D2: grazing with oat and alfalfa hay; D3: grazing with concentrated supplement. ALP: alkaline phosphatase; ALT: alanine aminotransferase; *p*-value BH: *p*-value adjusted by the Benjamini–Hochberg method. Averages of blood parameters with different superscripts (^a, b, c^) indicate statistically significant differences (*p* < 0.05).

**Table 5 vetsci-12-00687-t005:** Goats’ blood biochemistry by season, diet, and sampling time.

Season	Diet	Sampling	Urea	Protein	Albumin	ALP	ALT
			(mg/dL)	(g/dL)	(g/dL)	(U/L)	(U/L)
Rainy	D1	M1	44.73 ^c^	11.53 ^bc^	3.42 ^a^	260.60 ^de^	14.60 ^defg^
		M2	43.73 ^cd^	11.01 ^c^	2.83 ^bc^	334.73 ^cde^	15.73 ^de^
		M3	57.33 ^b^	13.19 ^a^	2.58 ^def^	338.93 ^cde^	17.60 ^cd^
	D2	M1	44.87 ^c^	11.23 ^c^	3.33 ^a^	599.53 ^abcde^	13.20 ^efg^
		M2	45.20 ^c^	10.71 ^cd^	2.49 ^ef^	597.27 ^abcde^	15.33 ^def^
		M3	58.73 ^b^	10.99 ^c^	2.48 ^ef^	459.57 ^bcde^	15.40 ^de^
	D3	M1	47.27 ^c^	11.39 ^bc^	3.42 ^a^	822.07 ^ab^	14.00 ^efg^
		M2	31.00 ^fg^	10.45 ^cde^	2.43 ^f^	755.53 ^abc^	11.93 ^g^
		M3	33.87 ^fg^	12.39 ^ab^	2.48 ^ef^	630.87 ^abcd^	15.27 ^def^
Dry	D1	M1	37.73 ^def^	9.50 ^ef^	2.81 ^bc^	143.07 ^e^	23.93 ^a^
		M2	27.33 ^g^	8.95 ^f^	2.67 ^cde^	343.60 ^cde^	14.00 ^efg^
		M3	34.64 ^ef^	11.01 ^c^	2.69 ^cd^	252.36 ^de^	12.87 ^efg^
	D2	M1	44.07 ^cd^	9.52 ^ef^	2.89 ^b^	955.73 ^a^	20.60 ^bc^
		M2	37.27 ^def^	9.28 ^f^	2.72 ^bcd^	737.87 ^abc^	14.33 ^efg^
		M3	31.18 ^fg^	10.93 ^c^	2.77 ^bcd^	766.15 ^abc^	12.07 ^fg^
	D3	M1	44.73 ^c^	9.79 ^def^	2.83 ^bc^	862.93 ^ab^	23.13 ^ab^
		M2	43.73 ^cd^	9.21 ^f^	2.71 ^bcd^	610.30 ^abcde^	15.27 ^def^
		M3	57.33 ^b^	10.87 ^cd^	2.68 ^cde^	788.45 ^abc^	13.27 ^efg^
Standard error			2.26	0.36	0.06	130.62	0.98

M1, M2, M3: Sampling 1, 2, and 3; 1: exclusive grazing; D2: grazing with oat and alfalfa hay; D3: grazing with concentrated supplement. ALP: alkaline phosphatase; ALT: alanine aminotransferase; The averages of blood parameters with different superscripts (^a, b, c, d, e, f, g^) indicate statistically significant differences (*p* < 0.05).

**Table 6 vetsci-12-00687-t006:** Hematology in goats by season and diet (values adjusted with p-value BH).

Season	Diet	Leukocytes	Neutrophils	Hemoglobin	MCV	Lymphocytes	Body Weight
		(Cells/µL)	(%)	(g/dL)	(fL)	(%)	(kg)
Rainy	D1	7.38 ^c^	43.12 ^a^	9.28 ^bc^	18.47 ^b^	39.88 ^a^	25.15 ^c^
	D2	8.52 ^c^	39.56 ^a^	10.50 ^a^	18.43 ^b^	42.72 ^a^	24.13 ^c^
	D3	8.28 ^c^	38.65 ^a^	9.66 ^b^	18.13 ^c^	45.92 ^a^	25.32 ^c^
Dry	D1	7.43 ^c^	40.67 ^a^	8.96 ^bc^	18.41 ^b^	40.10 ^a^	26.95 ^b^
	D2	18.89 ^a^	42.81 ^a^	8.74 ^c^	18.70 ^a^	40.64 ^a^	25.06 ^c^
	D3	13.61 ^b^	34.93 ^a^	8.92 ^bc^	18.40 ^b^	48.83 ^a^	29.05 ^a^
Standard error		1.44	2.40	0.21	0.07	2.79	0.51

D1: exclusive grazing; D2: grazing with oat and alfalfa hay; D3: grazing with concentrate supplement. MCV: mean corpuscular volume; MCHC: mean corpuscular hemoglobin concentration; BH: *p*-value adjusted by the Benjamini–Hochberg method. Averages of blood parameters with different superscripts (^a, b, c^) indicate statistically significant differences (*p* < 0.05).

**Table 7 vetsci-12-00687-t007:** Hematology in goats by season, diet, and sampling time.

Season	Diet	Sampling	Leukocytes	Neutrophils	Hemoglobin	MCV	Lymphocytes
			(Cells/µL)	(%)	(g/dL)	(fL)	(%)
Rainy	D1	M1	7.21 ^b^	43.43 ^a^	9.47 ^cdef^	18.25 ^bc^	41.03 ^a^
		M2	7.30 ^b^	38.10 ^a^	9.55 ^cdef^	18.46 ^bc^	43.63 ^a^
		M3	7.64 ^b^	47.83 ^a^	8.82 ^fgh^	18.70 ^b^	34.97 ^a^
	D2	M1	8.53 ^b^	44.89 ^a^	10.87 ^b^	18.35 ^bc^	38.81 ^a^
		M2	8.32 ^b^	32.96 ^a^	10.50 ^bc^	18.47 ^b^	48.53 ^a^
		M3	8.71 ^b^	40.84 ^a^	10.12 ^bcd^	18.46 ^bc^	40.83 ^a^
	D3	M1	7.21 ^b^	45.73 ^a^	9.80 ^bcdef^	18.10 ^bc^	41.69 ^a^
		M2	8.21 ^b^	32.92 ^a^	9.99 ^bcde^	18.11 ^bc^	50.45 ^a^
		M3	9.43 ^b^	37.30 ^a^	9.19 ^def^	18.19 ^bc^	45.62 ^a^
Dry	D1	M1	9.19 ^b^	44.97 ^a^	11.15 ^ab^	18.14 ^bc^	37.80 ^a^
		M2	5.80 ^b^	35.69 ^a^	6.78 ^ij^	18.50 ^b^	41.44 ^a^
		M3	7.31 ^b^	41.35 ^a^	8.94 ^efg^	18.59 ^b^	41.06 ^a^
	D2	M1	9.34 ^b^	43.63 ^a^	11.99 ^a^	18.19 ^bc^	38.49 ^a^
		M2	13.43 ^b^	41.58 ^a^	8.56 ^fgh^	18.73 ^b^	40.20 ^a^
		M3	33.88 ^a^	43.24 ^a^	5.67 ^j^	19.18 ^a^	43.22 ^a^
	D3	M1	10.52 ^b^	33.67 ^a^	11.08 ^ab^	18.05 ^c^	48.95 ^a^
		M2	15.48 ^b^	33.34 ^a^	7.80 ^hi^	18.70 ^b^	50.07 ^a^
		M3	14.83 ^b^	37.78 ^a^	7.87 ^ghi^	18.43 ^bc^	47.46 ^a^
Standard error			2.37	4.16	0.36	0.12	3.69

M1, M2, M3: Sampling at 30, 45, and 60 days after the beginning of the experiment. D1: exclusive grazing; D2: grazing with oat and alfalfa hay; D3: grazing with concentrate supplement. MCV: mean corpuscular volume; MCHC: mean corpuscular hemoglobin concentration. The averages of blood parameters with different superscripts (^a, b, c, d, e, f, g, h, i, j^) indicate statistically significant differences (*p* < 0.05).

## Data Availability

The information published in this study is available on request from the corresponding author.

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
