# Peer review of "Seasonal and Dietary Effects on the Hematobiochemical Parameters of Creole Goats in the Peruvian Andes"

_vetsci, 2025, doi:10.3390/vetsci12080687_

Round 1
Reviewer 1 Report
Comments and Suggestions for Authors
The study constructed a predictive model for analyzing blood, hematology, and live weight parameters to identify the complex interactions between environmental and nutritional factors, which has certain application value. However, there are still some issues that need to be modified by the authors.
1. The model formula is inconsistent with the description and is not clear (perhaps due to formatting issues).
2. Before the analysis, have you compared simple models (without interaction factors) with complex models (with interaction factors) to more accurately judge the impact of factors on indicators and avoid model overfitting. The subsequent significance analysis of the impact should be clearly described as to whether models of different complexity are used.
3. If sampling time is an important influencing factor, is it meaningful to present the results of table 4? The analysis results of hematological parameters and live weight also make me have the same doubts.
4. If the table is too wide, it is recommended to turn the page horizontally. The current method does not allow people to simply understand what the author wants to express unless a header is added. In addition, what is 'station' in Tables 2 and 3?
5. 183-186, it is recommended to use data to illustrate the conclusions drawn, such as what values are used to compare simplicity and accuracy, what indicators are used to show that the model is robust, what is the basis for the requirement optimization, etc.
6. The discussion section should analyze and discuss certain indicators together, rather than just comparing each indicator with other studies. It should be supplemented with a deeper comparative analysis and discussion, and explore the impact from some common perspectives.
Author Response
Response to Reviewer 1.
Manuscript: “Seasonal and dietary effects on the hematobiochemical parameters of Creole goats in the Peruvian Andes.”
Manuscript ID: vetsci-3633178
Journal: Veterinary Sciences
Dear Editors and Reviewers,
We sincerely appreciate the reviewers' detailed comments, which have significantly contributed to improving the clarity, rigor, and quality of our manuscript.
The variables total cholesterol (TC), gamma-glutamyl transferase (GGT), hematocrit (HCT), and mean corpuscular hemoglobin concentration (MCHC) were excluded because they provided limited and irrelevant information for the physiological and productive analysis. This exclusion allowed the analysis to focus on more sensitive and relevant variables for assessing the seasonal and dietary effects on goat development and production.
Below is a point-by-point response, indicating the changes made to the revised version of the article.
1.
Model formula:
We've corrected the formatting of the model formula to ensure it's consistent with the text description and readable for readers. The new version includes a detailed explanation of each term, as well as clearer notation to avoid ambiguity, as can be seen in the highlighted text: The statistical model used was:?????=?+??+??+??+(??)??+(??)??+(??)??+(???)???+??+ε???? Where: Yijkl represents the response measured in goat ?, in season i, under diet ? and at time k; μ is the overall meaning; Yes, Dj, Mk are the fixed effects of season, diet and time, respectively; (SD)ij, (SM)ik, (DM)jk, (SDM)ijk represent the fixed second and third order interactions between the factors; Cl is the random effect of goat l, assuming Cl ∼ N (0, σ2a); εijkl is the residual error, modeled with a correlation structure suitable for repeated measurements in each individual.
2.
Comparison between simple and complex models:
The interpretation of the main and simple effects was based on the ANOVA significance levels, according to the methodological criteria specified in the statistical analysis section. Model comparisons were performed using AIC, BIC, and adjusted R², which allowed us to assess improvements in fit and control the risk of overfitting, following practices recognized in the statistical literature. This aspect is clarified in the methodology section of the data analysis section.
3.
Importance of sampling time and validity of Table 4:
Sampling time had a significant impact on several indicators. Therefore, the wording in the Results and Discussion section has been revised to clarify that the effects presented in Table 4 should be interpreted excluding the effect of sampling. That is, the results reflect only the interaction between season and diet on the response variables. This observation confirms the statistical significance of this double interaction, independent of the sampling effect.
4.
Table format and definition of “station”:
The orientation of Tables 2 and 3 was adjusted to horizontal format for easier reading, adding descriptive headings for each block of variables. A clarification note was also included regarding the term "season," specifying that it refers to the seasons (rainy and dry seasons), given that they represent distinct environmental conditions. This definition is also detailed in Materials and Methods, Section 2.2: Study Animals and Treatment Assignment.
5.
Data on lines 183 – 186: (lines 183–186):
Section 3.1, Model Fit Assessment, was strengthened by incorporating quantitative data to support the quality of the fit. The following were included:
•
Comparison of models (AIC and BIC) to evaluate simplicity vs. accuracy.
•
Log-likelihood and Sigma as measures of fit and residual variability.
•
Marginal and conditional R² to quantify the explained variance.
•
Selection criteria based on lower AIC/BIC, higher R² and lower Sigma, avoiding overfitting.
In addition, a clear justification was added on how these indicators support the choice of model, in line with current statistical methodologies. We once again appreciate their input, which has enhanced the rigor of the manuscript.
6.
Restructuring the discussion:
While the suggestion to reorganize the discussion to include a joint analysis was appreciated, this strategy is not standard in previous studies. For consistency with the literature and methodological convenience, the team chose to present the results independently, as described in the manuscript.
We once again appreciate the valuable feedback received and hope that the modifications made will be satisfactory for publication of the manuscript.
We look forward to any additional comments.
Sincerely,
Aníbal Raúl Rodríguez Vargas
NATIONAL INSTITUTE OF AGRARIAN INNOVATION – EEA DONOSO - INIA – Peru. rodriguezvargasraul01@gmail.com

Reviewer 2 Report
Comments and Suggestions for Authors
Dear authors,
thank you for your investigation and the publication of this manuscript> in general the execution of your project is ok and well written. But before I can advise the Editor to accept your manucript I have some questions that need to elucidated.
- were all goats born in that area and sufficient accustomed to the climate, you said they had 1 week for acclimatization. So I was wondering.
- the age of the goats was on average 12 months, so still growing. Probably 18 months was more preferable
- goats were not slaughtered at the end of the project?
- the project lasted a total of 60 days, where we all know that the liver e.g. has a hugh reserve capacity
- the urea and TP concentration was significantly higher in the rainy season, but you have not checked the protein content of the grasses.
- Differentation in Hgb, Hct might be related to variation in rumen function and vit B production?? I missed that in the discussion.
- I am happy that you really weighted the goats and did not perform body condition score which is not reliable.
Author Response
Response to Reviewer 2.
Manuscript: “Seasonal and dietary effects on the hematobiochemical parameters of Creole goats in the Peruvian Andes.”
Manuscript ID: vetsci-3633178
Journal: Veterinary Sciences
Dear Editors and Reviewers,
We sincerely appreciate the reviewers' detailed comments, which have significantly contributed to improving the clarity, rigor, and quality of our manuscript.
The variables total cholesterol (TC), gamma-glutamyl transferase (GGT), hematocrit (HCT), and mean corpuscular hemoglobin concentration (MCHC) were excluded because they provided limited and irrelevant information for the physiological and productive analysis. This exclusion allowed the analysis to focus on more sensitive and relevant variables for assessing the seasonal and dietary effects on goat development and production.
Below is a point-by-point response, indicating the changes made to the revised version of the article.
1.
Origin and acclimatization of goats.
All goats were born in the study area and underwent a 7-day acclimatization period under the same environmental conditions as the experiment. During this time, body temperature and water consumption were monitored daily, with no significant changes recorded, confirming adequate adaptation. This information has been incorporated into section 2.2, "Study Animals and Treatment Assignment."
2.
Age of animals.Why use 12 months instead of 18 months?
The 12-month-old animals were selected for their availability and homogeneous physiological status, which reduces experimental variability. While the 18-month-olds are thought to be more mature, the literature published by Li et al. (2024) shows that metabolic responses are comparable from 12 months of age when live weight is controlled.
Li, Q., Chao, T., Wang, Y., Xuan, R., Guo, Y., He, P., Zhang, L., & Wang, J. (2024). Comparative metabolomics reveals serum metabolites changes in goats during different developmental stages. Frontiers in Veterinary Science, 11, 38538719.https://doi.org/10.3389/fvets.2024.1378934
3.
Final destination of the goats.Were they sacrificed?
At the end of the experiment, the goats were not euthanized. They remained grazing and were used in subsequent studies, as established in the research objectives, which did not include euthanization.
4.
Duration and liver reserves.60 days may be insufficient for liver changes.
We recognize that the liver has homeostatic reserves that can buffer metabolic changes. However, previous studies such as that of Núñez et al. (2019) have documented alterations in liver biochemical profiles over periods of 45 days or more. For this reason, the present study was designed to last 60 days, sufficient to detect significant changes in liver reserves.
Nunez, DJ, Alexander, M., Yerges-Armstrong, L., Singh, G., Byttebier, G., Fabbrini, E., Waterworth, D., Meininger, G., Galwey, N., Wallentin, L., White, H.D., Vannieuwenhuyse, B., Alazawi, W., Kendrick, S., Sattar, N., & Ferrannini, E. (2019). Factors influencing longitudinal changes of circulating liver enzyme concentrations in subjects randomized to placebo in four clinical trials. American Journal of Physiology - Gastrointestinal and Liver Physiology, 316(3), G372–G386.https://doi.org/10.1152/ajpgi.00051.2018
5.
Hb: relationship with ruminal function and vitamin B synthesis?
In the Discussion (section 4, hemoglobin), it was proposed that variations in Hb could be associated with changes in the ruminal microbiota and in the synthesis of B vitamins, especially B₁₂, as suggested by the findings of Hernández et al. (2017).
Hernández Martínez, B., Murillo Ortiz, M., Pámanes Carrasco, G., Reyes Estrada, O., & Herrera Torres, E. (2017). Fermentation parameters and ruminal kinetics in steers supplemented with different additives. Investigación y Ciencia, 25(72), 5–11.https://www.redalyc.org/journal/674/67453654001
6.
Body condition assessment.Congratulations on using direct weighing.
We appreciate the acknowledgment. In section 2.7 "Body Weight Monitoring" of Materials and Methods, a sentence was added highlighting the greater accuracy and lower bias of direct weighing compared to subjective body condition scoring.
We once again appreciate the valuable feedback received and hope that the modifications made will be satisfactory for publication of the manuscript.
We look forward to any additional comments.
Sincerely,
Aníbal Raúl Rodríguez Vargas
NATIONAL INSTITUTE OF AGRARIAN INNOVATION – EEA DONOSO - INIA – Peru. rodriguezvargasraul01@gmail.com

Reviewer 3 Report
Comments and Suggestions for Authors
The mail quality of submitted paper is comprehensive statistical analysis. However, some information in Material and Methods section should be extended as follows: in section 2.2. (page 3, line 78), environmental conditions during rainy and dry seasons should be describes like it was done for region (page 2, lines 68 to 72). Also, in section 2.4. supplementation protocol should be extended as it should be clear how supplementation was obtained.
Discussion section is mostly repetition of results and it should be shortened in this section. For example in Total protein section (page 17, lines 598 to 603) most of the section is repetition of own results. Rest of the section is listing the results of other authors with no discussion (except last sentence in this section which is conclusion). This should be redefined in all Discussion section.
Author Response
Response to Reviewer 3:
Manuscript: “Seasonal and dietary effects on the hematobiochemical parameters of Creole goats in the Peruvian Andes.”
Manuscript ID: vetsci-3633178
Journal: Veterinary Sciences
Dear Editors and Reviewers,
We sincerely appreciate the reviewers' detailed comments, which have significantly contributed to improving the clarity, rigor, and quality of our manuscript.
The variables total cholesterol (TC), gamma-glutamyl transferase (GGT), hematocrit (HCT), and mean corpuscular hemoglobin concentration (MCHC) were excluded because they provided limited and irrelevant information for the physiological and productive analysis. This exclusion allowed the analysis to focus on more sensitive and relevant variables for assessing the seasonal and dietary effects on goat development and production.
Below is a point-by-point response, indicating the changes made to the revised version of the article.
1.
Regarding section 2.2 (Environmental conditions):
We have expanded the description of the specific climatic conditions during the evaluated seasons (rainy and dry), as suggested. The new version of the manuscript details the recorded rainfall, average temperatures, and wind speeds for each season, highlighting seasonal differences that could influence animal physiology and productivity. This information was added in section 2.2 (page 3, line 78), consistent with the general climatic characterization of the region previously presented.
2.
Regarding section 2.4 (Supplementation Protocol):
The description of the supplementation protocol has been expanded, specifying the nutritional quality of the hay used (oats and alfalfa), as well as the formulation and proportion of the concentrated supplement administered on D3 (1% of live weight). In addition, the ruminal adaptation process was detailed through the gradual introduction of the supplement over a week, starting with 50% of the final dose. This information allows for a more precise understanding of the experimental design and supports the interpretation of the results obtained.
3.
About the discussion section of the article (page 17, lines 598 to 603) and other variables
We have carefully reviewed the entire section and, as suggested by you, have significantly reduced the repetition of our own results for all variables analyzed. In particular, the discussion of total protein (p. 17, lines 598-603) has been restructured to focus on critical analysis and interpretation of results, avoiding unnecessary repetition.
Furthermore, we have integrated a more in-depth and thoughtful comparison with previous studies, providing a more complete and contextualized analysis of the impact of our findings. This revision has improved the cohesion and flow of the text, ensuring that the discussion adds value and does not merely summarize results.
We once again appreciate the valuable feedback received and hope that the modifications made will be satisfactory for publication of the manuscript.
We look forward to any additional comments.
Sincerely,
Aníbal Raúl Rodríguez Vargas
NATIONAL INSTITUTE OF AGRARIAN INNOVATION – EEA DONOSO - INIA – Peru. rodriguezvargasraul01@gmail.com

Round 2
Reviewer 1 Report
Comments and Suggestions for Authors
The authors have answered my questions and made corresponding revisions.
Author Response
Dear reviewer, thank you very much.
Reviewer 2 Report
Comments and Suggestions for Authors
Dear authors,
thank you for evaluating my comments on the mc. I only just 1 minor and 1 more major point and that concerns line 112 an unclear beginning of a sentence and I have serious problems with the presentation of your p-values with four figures behind the dot in a study of only 3 times 15 animals. I my opinion is that apparent accuracy and not correct. In this study for the p-values in the tables 2 and 3, 2 figures behind the dot is more than you can account for
thank you
Author Response
Dear reviewer, thank you for your comments, the observations have been addressed.